# PIGN-Related Disease in Two Lithuanian Families: A Report of Two Novel Pathogenic Variants, Molecular and Clinical Characterisation

**DOI:** 10.3390/medicina58111526

**Published:** 2022-10-26

**Authors:** Evelina Siavrienė, Živilė Maldžienė, Violeta Mikštienė, Gunda Petraitytė, Tautvydas Rančelis, Justas Dapkūnas, Birutė Burnytė, Eglė Benušienė, Aušra Sasnauskienė, Jurgita Grikinienė, Eglė Griškevičiūtė, Algirdas Utkus, Eglė Preikšaitienė

**Affiliations:** 1Department of Human and Medical Genetics, Institute of Biomedical Sciences, Faculty of Medicine, Vilnius University, 08410 Vilnius, Lithuania; 2Department of Bioinformatics, Institute of Biotechnology, Life Sciences Center, Vilnius University, 10257 Vilnius, Lithuania; 3Centre for Medical Genetics, Vilnius University Hospital Santaros Klinikos, 08410 Vilnius, Lithuania; 4Department of Biochemistry and Molecular Biology, Institute of Biosciences, Life Sciences Centre, Vilnius University, 10257 Vilnius, Lithuania; 5Centre of Pediatrics, Institute of Biomedical Sciences, Faculty of Medicine, Vilnius University, 03101 Vilnius, Lithuania; 6Faculty of Medicine, Vilnius University, 03101 Vilnius, Lithuania

**Keywords:** *PIGN*, compound heterogeneous variants, multiple congenital anomalies-hypotonia-seizures syndrome 1, Fryns syndrome, congenital anomalies

## Abstract

*Background and Objectives*: Pathogenic variants of *PIGN* are a known cause of multiple congenital anomalies-hypotonia-seizures syndrome 1 (MCAHS1). Many affected individuals have clinical features overlapping with Fryns syndrome and are mainly characterised by developmental delay, congenital anomalies, hypotonia, seizures, and specific minor facial anomalies. This study investigates the clinical and molecular data of three individuals from two unrelated families, the clinical features of which were consistent with a diagnosis of MCAHS1. *Materials and Methods*: Next-generation sequencing (NGS) technology was used to identify the changes in the DNA sequence. Sanger sequencing of gDNA of probands and their parents was used for validation and segregation analysis. Bioinformatics tools were used to investigate the consequences of pathogenic or likely pathogenic *PIGN* variants at the protein sequence and structure level. *Results*: The analysis of NGS data and segregation analysis revealed a compound heterozygous NM_176787.5:c.[1942G>T];[1247_1251del] *PIGN* genotype in family 1 and NG_033144.1(NM_176787.5):c.[932T>G];[1674+1G>C] *PIGN* genotype in family 2. In silico, c.1942G>T (p.(Glu648Ter)), c.1247_1251del (p.(Glu416GlyfsTer22)), and c.1674+1G>C (p.(Glu525AspfsTer68)) variants are predicted to result in a premature termination codon that leads to truncated and functionally disrupted protein causing the phenotype of MCAHS1 in the affected individuals. *Conclusions*: PIGN-related disease represents a wide spectrum of phenotypic features, making clinical diagnosis inaccurate and complicated. The genetic testing of every individual with this phenotype provides new insights into the origin and development of the disease.

## 1. Introduction

Multiple congenital anomalies-hypotonia-seizures syndrome (MCAHS) is a group of genetically heterogeneous disorders characterised by multiple congenital anomalies, neonatal hypotonia, seizures, minor facial anomalies, and psychomotor development delay. Depending on the causative gene, four different subtypes of MCAHS have been identified to date [1,2]. Homozygous or compound heterozygous variants in the *PIGN* (Phosphatidylinositol Glycan Anchor Biosynthesis Class N; OMIM #606097) gene, found in seven individuals from related Arab Israeli families, have been first reported to result in MCAHS1 (OMIM #614080, ORPHA #280633) [3]. Genetic variants in the *PIGA* (OMIM #311770), *PIGT* (OMIM #610272), and *PIGQ* (OMIM #605754) genes were found to cause for MCAHS2 (OMIM #300868; ORPHA #300496), MCAHS3 (OMIM #615398; ORPHA #369837), and MCAHS4 (OMIM #618548), respectively [2]. According to the Human Gene Mutation Database (HGMD), pathogenic DNA sequence variants, the majority of which are missense or nonsense, most commonly occur in the *PIGN* gene; an autosomal recessive MCAHS1 is therefore the most prevalent form of MCAHS [4].

Some affected individuals with MCAHS1 have clinical features meeting the diagnostic criteria for Fryns syndrome (FS; MIM #229850; ORPHA #2059) [5,6]. The syndrome was first described by Fryns et al. (1979) when the authors reported two female siblings born with identical multiple congenital anomalies [7]. Based on published data, Lin et al. (2005) formulated six clinical diagnostic criteria for FS, which include diaphragm defects, characteristic facial appearance, distal digital hypoplasia, significant pulmonary hypoplasia, at least one characteristic anomaly (polyhydramnios, cloudy cornea, orofacial cleft, brain malformation, cardiovascular malformation, renal dysplasia, gastrointestinal malformation, or genital malformation), and an affected sibling. The presence of at least four criteria provides a narrow definition of FS, while the presence of three criteria fits the broader definition of FS [8]. To date, seventy-six individuals with PIGN-related disease have been described in scientific literature, but some of them do not fully meet the diagnostic criteria of both MCAHS1 and FS [9,10,11]. Therefore, the understanding of genotypic–phenotypic correlation in PIGN-related disorders needs to be improved.

In this study, we unravel two previously unpublished DNA sequence variants of *PIGN* and provide novel insights about recurrent variants, the consequences, and the possible pathogenesis mechanism of which were evaluated by detailed molecular and in silico analysis. In this regard, we define the genotype-phenotype correlation in two premature siblings and one unrelated female individual from the Lithuanian population who are affected by PIGN-related disease.

## 2. Materials and Methods

### 2.1. Clinical Evaluation of the Individuals

In the present study, we describe three individuals with different clinical manifestations of PIGN-related disorder in two unrelated Lithuanian families (Appendix A).

Family 1. A male foetus from the second pregnancy (F1.II-2) was born to non-consanguineous parents. At 28 weeks of gestation, a diaphragmatic hernia was detected by an ultrasound examination. Deformity of the left hand (fixed through the wrist joint), clubfoot, deformed toes on one foot, micropenis, cryptorchidism, craniosynostosis, ventriculomegaly, and minor facial anomalies were observed, including widely spaced eyes, wide nasal bridge, short nose, wide mouth, downturned corners of mouth, and prominent supraorbital ridges. Array comparative genomic hybridisation using the DNA of uncultured amniocytes revealed no potentially pathogenic genomic structural abnormalities. The foetus was born at 33 gestational weeks in spontaneous labour but died of progressive cardiopulmonary failure after surviving 22 min. Morphological examination revealed tetralogy of Fallot (high ventricular septal defect, aortic dextroposition, pulmonary stenosis, right ventricular hypertrophy), bilateral congenital diaphragmatic hernia (CDH) with abdominal organ eventration into the chest, and pulmonary hypoplasia, craniosynostosis, minor ventriculomegaly, lobulated spleen, the irregular position of the left wrist and foot, the cleft hard palate, and other facial anomalies (protruding forehead, broad nasal bridge, and widely spaced eyes).

At 24 gestational weeks during the fourth pregnancy, a prenatal ultrasound showed polyhydramnios and a female foetus (F1.II-4) with multiple developmental defects, including CDH, cleft palate, a congenital cardiac malformation (ventricular septal defect with dextroposition of the heart), and minor facial anomalies, including wide nasal bridge, broad nasal tip, wide mouth, low-set ears, and absent nasal bone (Figure 1A–D). Array comparative genomic hybridisation using DNA of uncultured amniocytes showed normal results. The birth started naturally at 35 gestational weeks. The birth weight of the newborn was 3300 g (90–97th centile), her length was 49 cm (75th centile), and her head circumference was 35 cm (90–97th centile). Apgar scores at 1 and 5 min were 6 and 8 respectively. The newborn was intubated and ventilated immediately after birth and transferred to the neonatal intensive care unit. The neonate’s condition was critical due to respiratory failure and prematurity. She had a cleft hard palate, CDH, and minor anomalies of the face, hands, and feet. Physical examination revealed bluish-white cyanotic skin, prominent forehead, widely spaced eyes, broad nasal bridge, and nasal tip, periorbital fullness, deep and short philtrum, wide mouth, cleft palate, low-set ears, hypoplastic earlobe, short neck, nuchal crease, hypoplastic nails on both hands and feet, and sacral dimple. Neurosonography showed slightly enlarged lateral ventricles (right more than left), periventricularly increased parenchymal echogenicity, fragmented vascular plexus on the right, and enlarged 3rd ventricle. Renal ultrasound examination revealed bilateral hydronephrosis. Echocardiogram identified perimembranous ventricular septal defect with bilateral shunt, markedly reduced left ventricular inotropy, patent ductus arteriosus, and dextrocardia (due to CDH). The new-born’s condition was extremely critical and deteriorated rapidly. Asystole developed after loss of vital functions 14 h after birth.

Family 2. The proband (F2.II-2), a 22-month-old female, was born to healthy non-consanguineous parents. She was born by urgent Caesarean section at 39 gestational weeks due to breech presentation. Her birth weight was 4007 g (97th centile), her height was 59 cm (>97th centile), and her head circumference was 37 cm (>97th centile). Apgar score was 9 at 1 min and 9 at 5 min after birth. During the first months of life, the proband had central hypotonia, gross motor delay, and slow emotions.

Seizures similar to infantile spasms have been observed since her first months. At the age of 7 months, epilepsy–West syndrome was diagnosed. The initial electroencephalogram showed hypsarrhythmia. Brain magnetic resonance imaging revealed only a thin splenium of corpus callosum. Neurodevelopment assessment at the age of 7 months showed generalised hypotonia with normal tendon reflexes. Global neurodevelopmental delay was noted due to sluggish psychoemotional reactions, her inability to turn from her back onto her abdomen, and her inability to sit up. Several antiepileptic drugs were tried (adrenocorticotropic hormone, vigabatrin, nitrazepam, and levetiracetam), but the effect was only partial. A ketogenic diet with a 3:1 ketogenic ratio was started after 6 weeks. Cannabidiol was then added to levetiracetam and spasm cessation was achieved at the age of 26 months. At her last follow-up at the age of 1 year 10 months, neurological examination further revealed strabismus, axial hypotonia, and global development delay in the absence of progress. She had a few minor facial anomalies including long eyelashes and a tented upper lip. An ophthalmological examination showed convergent alternant strabismus. A hearing investigation by otoacoustic emission testing revealed nothing abnormal.

### 2.2. DNA Extraction

Genomic DNA (gDNA) was isolated from peripheral blood leukocytes of the probands (F1.II-2, F1.II-4, F2.II-2) and their parents (F1.I-1, F1.I-2, F2.I-1, F2.I-2) using phenol–chloroform–isoamyl alcohol extraction method [12].

### 2.3. Next-Generation Sequencing

Family 1. Whole exome sequencing (WES) using the high-throughput next-generation Illumina (Illumina, Inc., San Diego, CA, USA) platform was used to sequence the samples of the F1.II-2 and F1.II-4. DNA libraries were generated using TruSeq Rapid Exome Library Prep kit (8 × 3plex) (Illumina, Inc., San Diego, CA, USA). In order to precisely measure the concentration of DNA libraries, Qubit dsDNA BR Assay kit (Thermo Fisher Scientific, Waltham, MA, USA) and Qubit fluorimeter (Thermo Fisher Scientific, Waltham, MA, USA) were used. Clusters were amplificated using cBot system (Illumina, Inc., San Diego, CA, USA), TruSeq PE Cluster Kit v3-HS (Illumina, Inc., San Diego, CA, USA), and TruSeq Dual Index Sequencing Primer Box–Paired End (Illumina, Inc., San Diego, CA, USA). WES was performed using TruSeq SBS Kit v3-HS (Illumina, Inc., San Diego, CA, USA) and by employing the HiScanSQ the HiScanSQ (Illumina, Inc., San Diego, CA, USA) genetic analyser.

Family 2. Targeted amplicon next-generation sequencing (NGS) strategy was applied to identify pathogenic variants related to the epilepsy of the proband F2.II-2. The custom Ion Torrent (Ion Personal Genome Machine; Thermo Fisher Scientific, USA) panel capturing the 300 targeted epilepsy genes (Appendix A) was designed with Ampliseq Designer software (Thermo Fisher Scientific, USA). Enrichment of exonic sequences was performed with an Ion AmpliSeq Library Kit 2.0 (Thermo Fisher Scientific, USA) and sequenced on an Ion PGM (Thermo Fisher Scientific, USA) using Ion PGM Hi-Q View Sequencing Kit and 318 Chip (Thermo Fisher Scientific, USA) according to the manufacturer’s protocol. Mapping and variants calling were performed using the Ion Torrent suite software v5.12.3 (Thermo Fisher Scientific, USA).

In the case of both WES and targeted NGS, the analysis of high-throughput sequencing data was started with alignment against The Human NCBI Build GRCh37 (hg19/2009) reference genome. The data annotation was made using the ANNOVAR v.2018Apr16 program [13]. Criteria provided by the American College of Human Genetics and Genomics (ACMG) [14], in silico tools and databases provided by ANNOVAR program (e.g., SIFT, Polyphen2, GERP++, CADD, ExAC, GnomAD, 1000 Genome Project data, NCBI dbSNP, NCBI ClinVar), and the relevant scientific literature were used to assess the pathogenicity of detected variants. After the filtering process, the candidate genome variants were checked and validated by the Integrative Genomics Viewer (IGV) visualization tool [15]. Additionally, in the case of family 1, VarAFT software [16] was used to analyse and compare whole family data. After the filtering process, the compound heterozygous variants of the *PIGN* gene were among the final filtered candidate variants.

### 2.4. Sanger Sequencing

gDNA samples were used for the following validation and segregation analysis, which was performed by Sanger sequencing. Polymerase chain reactions (PCR) of gDNA sequences flanking the pathogenic or likely pathogenic variants of the *PIGN* gene were performed using specific primers (Appendix A) designed with the Primer Blast tool. PCR was performed using Phusion High-Fidelity PCR Master Mix (Thermo Fisher Scientific, USA). PCR products were fractioned by 1.5% agarose gel (TopVision, Thermo Fisher Scientific, USA) electrophoresis and visualized under ultraviolet light. The PCR products were sequenced with BigDye^®^ Terminator v3.1 Cycle Sequencing Kit (Thermo Fisher Scientific, USA) and ABI 3130xL Genetic Analyser (Thermo Fisher Scientific, USA). The resulting sequences were aligned with the reference sequence of *PIGN* (NCBI: NG_033144.1; NM_176787.5) containing 31 exons.

### 2.5. Western Blot

A primary fibroblast cell line was obtained from the skin biopsy of probands’ (F1.II-2 and F1.II-4) parents. The growing cell line was cultured in AmnioMax and C-100 Basal Medium (Thermo Fisher Scientific, USA), which was supplemented with AmnioMAX C-100 Supplement (Thermo Fisher Scientific, USA) and Amphotericin B (Gibco, Waltham, MA, USA) according to the standard laboratory procedures for human cell cultures.

The lysis of detached fibroblasts was performed on ice using 1 mL RIPA buffer (Sigma Aldrich, Saint Louis, MO, USA) for 107 cells and the appropriate amount of Halt™ Protease Inhibitor Cocktail (Thermo Fisher Scientific, USA). Incubated cell lysates were centrifuged, and protein concentration was determined using a Pierce™ BCA Protein Assay Kit (Thermo Fisher Scientific, USA). To fraction protein samples, 10% SDS-PAGE was used at 120 V. Proteins were transferred to nitrocellulose membrane (Thermo Fisher Scientific, USA) by semidry blotting. Blots were probed with anti-PIGN antibody (ratio 1:2000; PA5-59149, Invitrogen, Thermo Fisher Scientific, USA) and with anti-β-actin antibody (ratio 1:5000, ab8227, Abcam, UK) for detection of β-actin as a loading control. Membrane-bound primary antibodies of PIGN and β-actin were detected using horseradish-peroxidase-conjugated secondary anti-rabbit antibody (31460, Thermo Fisher Scientific, USA) and chemiluminescence reagent Pierce™ ECL Western Blotting Substrate (Thermo Fisher Scientific, USA).

### 2.6. Post-Sequencing Data Analysis

The prediction of possible splice site alterations was performed using the SpliceAI tool [17]. Sequences of evolutionary distinct randomly selected species were obtained from the Ensembl genome browser [18], while a sequence alignment was produced using the Clustal Omega tool [19]. To assess possible changes in PIGN (UniProtKB: O95427) amino acid sequence, bioinformatic analysis at the protein level was performed using ExPASy Bioinformatics Resource Portal tool [20], Pfam 32.0 database [21], and UniProt database [22]. The structure of the PIGN protein was downloaded from the AlphaFold Database [23]. The structure of aberrant protein affected by variant p.(Leu311Trp) was modelled in the COMER web server using MODELLER tool and the AlphaFold model as template [24,25]. EMBL-EBI EMBOSS service was used for protein sequence alignment and analysis [26]. Homologous eukaryotic proteins were identified in the reference proteomes using the HMMER web server [27], and residue conservation was analysed using WebLogo [28]. The orientation of protein in the membrane was visualized using the PPM server [29]. Protein structures were visualized using PyMOL Molecular Graphics System (Version 2.5, Schrödinger, LLC).

## 3. Results

Clinical phenotype, age of presentation, and disease course of all the probands was consistent with a diagnosis of MCAHS1. The clinical features are reported for each individual in the text above and in Appendix A.

The NGS strategy was applied to detect disease-causing variants. The results of WES and segregation analysis data revealed compound heterozygous *PIGN* genotype NM_176787.5:c.[1942G>T];[1247_1251del], NP_789744.1:p.[Glu648Ter];[Glu416GlyfsTer22] in family 1, while targeted NGS followed by Sanger sequencing of the parents determined compound heterozygous *PIGN* genotype NG_033144.1(NM_176787.5):c.[932T>G];[1674+1G>C], NP_789744.1:p.[Leu311Trp];[?] in family 2. Any other potentially causative DNA sequence variants were not identified.

According to ClinVar database [30], c.1247_1251del (ID #871597; rs1444518753), c.932T>G variant (ID #426983; rs746882521), and c.1674+1G>C (ID #264640; rs376355678) variant is classified as pathogenic or likely pathogenic. Meanwhile, the nonsense c.1942G>T variant is not recorded in ClinVar [30], HGMD [31], or other databases. Both DNA sequence variants identified in family 1 have not been previously reported in the literature. Missense and splicing variants of family 2 have been reported in several studies [6,9,32,33,34]. In silico analysis with SpliceAI predicted that the splice site c.1674+1G>C variant would most probably affect pre-mRNA splicing.

Sanger sequencing revealed the origin of these variants in the families. In family 1, both siblings (F1.II-2 and F1.II-4) inherited c.1247_1251del variant from their father and c.1942G>T variant from their mother (Figure 1E). In the family 2, the father was determined to be a heterozygous carrier of the c.1674+1G>C splice site variant, while the mother was heterozygous for the c.932T>G variant (Figure 1F).

Western blot method was applied to evaluate the expression of wild-type PIGN and to investigate the presence of PIGN residue in fibroblast cell lines from the probands’ (F1.II-2 and F1.II-4) mother and father carrying one of the protein-truncating variants. We were not able to identify bands specific to PIGN–it resulted in several bands throughout the membrane with no differences between carriers’ and control cells.

At the protein level, the nonsense c.1942G>T variant and deletion c.1247_1251del cause the premature termination of the PIGN protein: NP_789744.1:p.(Glu648Ter) and NP_789744.1:p.(Glu416GlyfsTer22), respectively. The consequences of the splice site variant c.1674+1G>C on messenger RNA (mRNA) have been previously investigated by McInerney-Leo et al. (2016). The analysis of cDNA revealed that the donor splice site variant causes the skipping of exon 18, and consequently activates a cryptic splice site 55 nucleotides downstream of exon 19 NP_789744.1:p.(Glu525AspfsTer68) [6]. Therefore, a possible impact on the PIGN protein of these three loss-of-function variants is a truncated and functionally disturbed protein (Figure 2C). Meanwhile, the c.932T>G variant at the protein level replaces leucine with tryptophan at codon 311 NP_789744.1:p.(Leu311Trp) (Figure 2B). Sequence alignment of the PIGN protein in seven randomly selected species revealed that the regions encoded by the analysed c.1247_1251del (p.(Glu416GlyfsTer22)), c.1674+1G>C (p.(Glu525AspfsTer68)), and c.932T>G (p.(Leu311Trp)) variants are highly conserved, while the nonsense c.1942G>T (p.(Glu648Ter)) variant occurs in the non-conservative locus (Figure 2A, Appendix A).

According to the ACMG guidelines [14], the c.1942G>T and c.1247_1251del (rs1444518753) variants identified in the family 1 were classified as likely pathogenic (PVS1, PM2, and PP3 criteria). The genetic alterations c.932T>G (rs746882521) and c.1674+1G>C (rs376355678) determined in family 2 were classified as likely pathogenic (PM1, PM2, PP3, PP5, BP1) and pathogenic (PVS1, PM2, PP3, PP5) respectively.

## 4. Discussion

The PIGN-related disease is a rare Mendelian disorder caused by pathogenic or likely pathogenic variants in *PIGN*. To our knowledge, *PIGN*-altering variants have been reported in 33 males and 43 females from 66 unrelated families with a wide range of clinical manifestations. The data from several previous publications suggest that congenital anomalies should not be considered one of the main features of PIGN-related disease [11,32,35,36,37]. Thiffault et al. (2017) reported on a male who exhibited intellectual disability, hypotonia, and seizures without congenital anomalies or any obvious dysmorphic features [11]. However, minor facial anomalies, defects of the gastrointestinal and genitourinary systems, tremors, and nystagmus may also be observed in individuals with a milder phenotype (Appendix A). In contrast, individuals with a severe phenotype often present with CDH and cleft lip or palate. More severe facial anomalies, polyhydramnios, and cardiovascular, urogenital, and central nervous system malformations may also be identified in affected individuals [38]. The reasons for such a wide phenotypic range remain largely unknown. The manifestation of the disease might depend on the nature and localisation of the genetic alteration. Phenotype–genotype correlation analysis of 76 affected individuals showed that the majority of *PIGN*-altering variants are missense, but there are also many splicing and nonsense variants. About 70% of reported DNA sequence variants have been found in the compound heterozygous state of *PIGN* (Appendix A).

The *PIGN* gene, located at chromosome cytoband 18q21.33, encodes glycosylphosphatidylinositol (GPI) ethanolamine phosphate (EtNP) transferase 1 (UniProtKB: O95427), which is expressed in the endoplasmic reticulum. The experimental structure of this protein has not been solved yet, but a highly accurate model by AlphaFold [39] is available. The PIGN protein is 931 amino acids in length and is composed of two parts: globular lumenal and transmembrane domains [40]. The latter domain is comprised of an N-terminal helix and a C-terminal region (Figure 2B). Meanwhile, the globular domain, which belongs to the alkaline phosphatase superfamily, catalyses the transfer of the EtNP side branch to the first mannose of the GPI-anchor. Computational analysis of known amino acid changing variants showed that these variants are mostly located in the catalytic region of the PIGN protein. Variants in the transmembrane region are rare and mostly concentrated on the boundary between water and lipid bilayer. Furthermore, some of the mutated residues are conservative and likely belong to the active site of the enzyme (Appendix A).

Functionally, the PIGN protein is involved in the GPI-anchor biosynthesis pathway. The activity of more than 26 other genes (e.g., *PIGA*, *PIGT*, *PIGQ*) is needed for this process as well. The active protein complex anchors different proteins to the outer layer of the plasma membrane and post-translationally modifies them. At least 150 different proteins that are involved in a number of different processes such as embryonic development, signal transduction, cell adhesion, and immune response require GPI anchoring in order to be expressed on the cellular surface. Therefore, any alterations in *PIGN*, which is abundantly expressed in almost all human tissues, not only have an impact on the gene product itself, but also influence GPI-anchored proteins [38,41]. For example, Maydan et al. (2011) found that pathogenic *PIGN* variants lead to 10-fold reduced expression of GPI-anchored proteins (i.e., CD59) on the surface of fibroblasts in individuals with MCAHS1 syndrome [3]. This mechanism is supported by studies of a mouse model, which resulted in altered localisation of GPI-anchored proteins, defective signal transduction, and a holoprosencephaly-like phenotype [42,43]. Moreover, the most recent study has revealed that GPI biosynthesis may be severely altered by SARS-CoV-2 replication, thus indicating *PIGN* as a novel risk gene for COVID-19 [44]. Additionally, Teye et al. (2021) discovered the association of *PIGN* with the suppression of chromosomal instability in leukemic cells [45].

In the present study, we describe compound heterozygous *PIGN* genotypes in two unrelated Lithuanian families. A previously unpublished compound heterozygous NM_176787.5:c.[1942G>T];[1247_1251del] *PIGN* genotype was detected by WES and segregation analysis in two siblings (F1.II-2 and F1.II-4) of family 1. Additionally, missense c.932T>G and splice site c.1674+1G>C variant were detected in compound heterozygous state by the analysis of targeted NGS data in the proband F2.II-2 of family 2 (Figure 1E,F). In silico analysis predicted these variants to be pathogenic or likely pathogenic and most probably affecting the PIGN protein structure.

Sequence alignment of the PIGN protein across seven distant species in our study revealed that the regions encoded by the c.1247_1251del (p.(Glu416GlyfsTer22)), c.1674+1G>C (p.(Glu525AspfsTer68)), and c.932T>G (p.(Leu311Trp)) variants are highly conserved, thus suggesting their fundamental importance for living organisms. Even though the c.1942G>T (p.(Glu648Ter)) variant occurs in a non-conservative region (Figure 2A), this genetic alteration triggers a stop codon rather than a codon specifying a new amino acid, thus affecting downstream functional protein domains. Additionally, c.1247_1251del (p.(Glu416GlyfsTer22)) and c.1674+1G>C (p.(Glu525AspfsTer68)) variants are predicted to cause a premature termination codon (Figure 2C). Therefore, these three variants result either in protein truncation or mRNA degradation due to nonsense-mediated decay. Both reductions of PIGN mRNA and truncation of protein would be deleterious and contribute to the pathogenesis of PIGN-related disease. Western blot method was used to investigate the expression of wild type and truncated protein PIGN in fibroblasts derived from the parents of family 1. This experiment resulted in the identification of several bands of various lengths. We could not reliably indicate which band corresponds to the full length and to the truncated PIGN proteins (in case the protein residues would not be targets for protein degradation). Thus, for future experiments it would be beneficial to use small interfering RNA to silence the expression of PIGN and identify the bands in Western blots which correspond to PIGN protein—it would be the bands whose intensity would be reduced after the silencing.

Differently, the missense c.932T>G variant (p.(Leu311Trp)) found in family 2 does not affect protein expression, but it has been previously suggested that it could reduce enzymatic activity [34]. Since human PIGN and its homologs have not been studied in detail using biochemical methods and the active site residues have not been determined yet, they can only be inferred from residue conservation analysis among alkaline phosphatase superfamily proteins [46]. The mutated Leu311 residue is distant from the putative active site, which contains Glu52, Ser94, Glu214, His218, Glu262, His263, and His271 residues, and there is no contact with the membrane region (Figure 2B). Even though the Leu311 residue is highly conserved among the eukaryotic homologs and belongs to a conserved WxL motif (Appendix A), the significance of this motif remains unknown. Taking into account that PIGN interacts with multiple proteins, it can be speculated that this motif is necessary for some kind of protein–protein interactions. To establish the exact influence on enzymatic activity, a more detailed functional analysis is needed.

The novel nonsense and frameshift variants presumably resulting in loss-of-function PIGN protein were detected in two siblings (F1.II-2, F1.II-4), whose clinical features meet the narrow diagnostic criteria for FS. Both had characteristic facial appearance, defects in the cardiovascular and respiratory systems, CDH, and digital hypoplasia. Severe polyhydramnios was also present in both pregnancies. Due to these severe congenital anomalies, both newborns died shortly after birth. Previously, pathogenic or likely pathogenic *PIGN* variants have been identified in eight individuals with CDH and various congenital anomalies, which led to premature death in the majority of the reported cases [5,6,38,47]. The severity of the manifestation of the disease in these cases can presumably be explained by biallelic loss-of-function *PIGN* variants. Contrastingly, the missense c.932T>G and splicing c.1674+1G>C variant of the *PIGN* gene identified in the proband of family 2 (F2.II-2) led to a less severe phenotype. Notably, the c.932T>G variant is the most commonly reported *PIGN* disease causative variant, which previously was detected in 13 of 76 individuals. All reported individuals with this missense variant were compound heterozygous with a different second variant (Appendix A) [9,32,33,34]. Bayat et al. (2022) reported the same compound heterozygous NG_033144.1(NM_176787.5):c.[932T>G];[1674+1G>C] *PIGN* genotype in a female 6.5 years of age who demonstrated similar global developmental delay, seizures, and hypotonia. A more precise genotype–phenotype correlation is however not available due to the lack of detailed clinical information concerning the published individual [9]. Due to the fact that the missense c.932T>G variant does not truncate the PIGN protein, the residual protein function could explain a milder phenotype manifesting without major congenital malformations.

## 5. Conclusions

The genotype–phenotype relationship we observed in this study correlates with previously published data of individuals with pathogenic or likely pathogenic PIGN variants. Our findings provide further proof that biallelic loss-of-function variants in PIGN are likely to be associated with a severe disease phenotype. The clinical severity of the syndrome is therefore presumably related to the level of protein truncation. The scientific community and clinicians require more data and each individual with the expressed phenotype provides new insights into the understanding of this hereditary disorder.

## Figures and Tables

**Figure 1 medicina-58-01526-f001:**
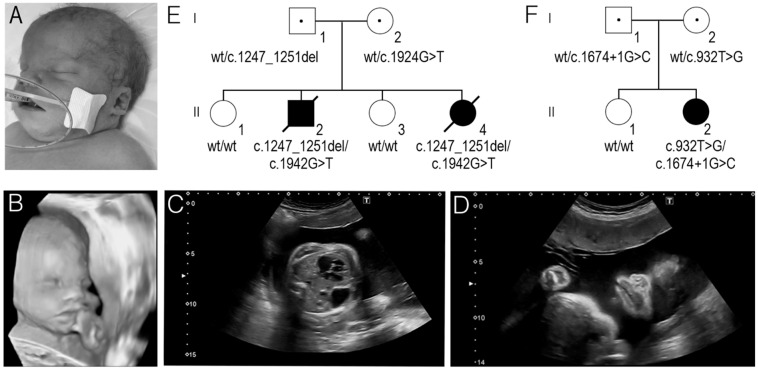
(**A**) The facial view of the second affected newborn in family 1 (F1.II-4). (**B**–**D**). A 3D ultrasound image of the second foetus in family 1 (F1.II-4). Note: wide nasal bridge, broad nasal tip, wide mouth, short philtrum (**B**); Congenital diaphragmatic hernia (**C**); Wide mouth (**D**). (**E**) The segregation of the c.1942G>T and c.1247_1251del *PIGN* variants in family 1. (**F**) The segregation of the c.932T>G and c.1674+1G>C *PIGN* variants in family 2.

**Figure 2 medicina-58-01526-f002:**
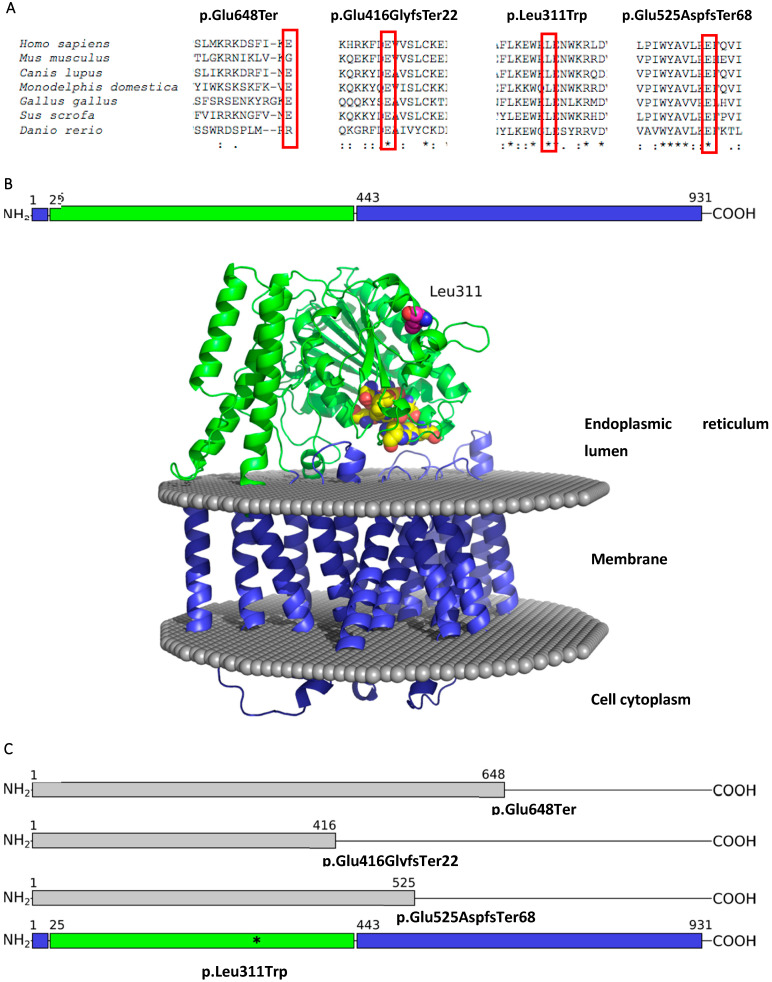
(**A**) Multiple sequence alignment produced by ClustalO of the PIGN protein across seven evolutionarily distant species. Below the amino acid sequences is a key denoting conserved sequence (*), conservative variants (:), semi-conservative variants (.), and non-conservative variants (); (**B**) The domain organization and structure model of human PIGN protein. The membrane region is coloured in blue, the lumenal domain is green, mutated residue Leu311 is shown in magenta coloured spheres, conserved residues that are likely to belong to enzyme active site are shown in yellow spheres; (**C**) Pathogenic or likely pathogenic variants visualized on the protein sequence.

## Data Availability

The main data generated and analysed during this study are included in this article and its Appendix A. Any additional information is available from the authors upon request.

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
