# Peer review of "PIGN-Related Disease in Two Lithuanian Families: A Report of Two Novel Pathogenic Variants, Molecular and Clinical Characterisation"

_medicina, 2022, doi:10.3390/medicina58111526_

Round 1
Reviewer 1 Report
The authors performed high-throughput sequencing of three individuals with different clinical manifestations of PIGN-related disorder (Multiple congenital anomalies-hypotonia-seizures syndrome, MCAHS) in two unrelated Lithuanian families. They reviewed the literature, and performed a little bit of protein modeling in an attempt to better define the genotype-phenotype correlation of PIGN pathogenic/likely pathogenic variants. Overall, the authors conducted a well designed study and provided a clear interpretation of the results. I believe this study is a good contribution to the rare diseases scientific community. Therefore, I recommend the acceptance of the manuscript. In regards to English language, minor spell check and grammar are required.
Author Response
We would like to thank this Reviewer for the review and positive evaluation. English language has been checked by expert before submission.
Reviewer 2 Report
The manuscript clearly presents the background, methodology followed, the results found and the conclusions as supported by the tools used to analyze the detected variants.
The authors also recognize the limitations of their study, and the fact that for some variants, further functional studies are necessary.
The authors honestly report on the unsuccessful results of the western blot, but rather than a short explanation on this in the Results Section, this needs to be further discussed and analyzed in the Discussion Section.
Author Response
We would like to thank Reviewer 2 for the observant review and comments. Regarding the Western blot results, we tried to clarify by including a brief discussion and suggestions on our results in the Discussion section lines 300–306. We also tried to clarify the information in the Results section lines 219–222.
Reviewer 3 Report
I thought that this paper was well written and informative. Plus aided by easily interpretable high solution images including the supplementary. Publishing more data on the detection of their variants in the 3 patients will aid the scientific community. I recommend the following corrections:
1. Introduction – Include the gene name (PIGN) in full, number of exons and cytogenic location.
2. Line 124. include the manufacturer of the Taq DNA Polymerase used for PCR
3. Line 156 – include what percentage agarose gel used.
4. Line 170 – include what antibody ratios you used e.g. 1:100 or 1:500
5. Supplementary table 3. Include the annealing temperature for the PCR amplification of PIGN
6. Lines 189-201 – Correct font style.
7. Line 213. “We were not able to get clear and reliable results possibly due to the used antibody not being tested in Western blot method earlier.” This statement needs to be reworded as unclear as why they were not tested earlier. I’m assuming that your antibody did not work, which is incredibly frustrating and that leads to the expensive task of seeking an alternative supplier (if available). I note that the brand of primary antibody only showed data for IHC/ICC and not an image of a western blot to shown expected band size(s). Perhaps, the protein of the expect size could be excised from the gel and sent for mass spectrophotometry for alternative analysis. Thus, I recommend that the western blot data be removed from the manuscript. Make a recommendation in the discussion about western blot.
Author Response
We would like to thank this Reviewer for the review and comments. We apologize for some inattentive mistakes and missing information, which has been corrected according to the comments below:
- Introduction – Include the gene name (PIGN) in full, number of exons and cytogenic location.
Answer: Full gene name has been indicated in the text (line 51); cytogenetic location has been indicated in the Discussion Section (line 261); number of exons has been indicated in the 2.4 Sanger sequencing section (line 166).
- Line 124. include the manufacturer of the Taq DNA Polymerase used for PCR
Answer: In our study we used Phusion DNA polymerase, which is included in Phusion High-Fidelity PCR Master Mix. This information together with manufacturer has been added into the text (line 162).
- Line 156 – include what percentage agarose gel used.
Answer: The percentage of agarose gel has been indicated in the text (line 162).
- Line 170 – include what antibody ratios you used e.g. 1:100 or 1:500
Answer: The ratios of antibodies that we used: for anti-β-actin antibody the ratio is 1:5000, and for anti-PIGN antibody the ratio is 1:2000. We included the ratios in the Method section lines 177–178.
- Supplementary table 3. Include the annealing temperature for the PCR amplification of PIGN
Answer: Annealing temperature has been added into the Supplementary table 3.
- Lines 189-201 – Correct font style.
Answer: The font style has been corrected.
- Line 213. “We were not able to get clear and reliable results possibly due to the used antibody not being tested in Western blot method earlier.” This statement needs to be reworded as unclear as why they were not tested earlier. I’m assuming that your antibody did not work, which is incredibly frustrating and that leads to the expensive task of seeking an alternative supplier (if available). I note that the brand of primary antibody only showed data for IHC/ICC and not an image of a western blot to shown expected band size(s). Perhaps, the protein of the expect size could be excised from the gel and sent for mass spectrophotometry for alternative analysis. Thus, I recommend that the western blot data be removed from the manuscript. Make a recommendation in the discussion about western blot.
Answer: We are truly thankful for Reviewers’ 3 insightful observation and suggestion. We sincerely apologise for the written sentence being unclear, difficult, or misleading. The results of the experiment using Western blot method for protein samples extracted from the parents’ of family 1 fibroblasts and controls showed several bands of various length with no differences between controls and parents. We were not reliably sure which band corresponds to the full length and to the truncated PIGN proteins (if the truncated proteins would not be directed to degradation). This is the reason why we indicated these results as unclear, insignificant. With respect to other Reviewers comments and recommendations, we tried to include a small discussion about these results and our suggestions in the Discussion section, lines 300–306. We also tried to clarify the information in the Results section lines 219–222. We hope that the added information describes our results and concerns clearer. The text included in Discussion section:
“Western blot method was used to investigate the expression of wild type and truncated protein PIGN in fibroblasts derived from the parents of family 1. This experiment resulted in identification of several bands of various length. We could not reliably indicate which band corresponds to the full length and to the truncated PIGN proteins (in case the protein residues would not be targets for protein degradation). Thus, for future experiments it would be beneficial to use small interfering RNA (siRNA) to silence the expression of PIGN and identify the bands in Western blots which correspond to PIGN protein – it would be the bands whose intensity would be reduced after the silencing. “